# Behavioral Responses of *Thrips hawaiiensis* (Thysanoptera: Thripidae) to Volatile Compounds Identified from *Gardenia jasminoides* Ellis (Gentianales: Rubiaceae)

**DOI:** 10.3390/insects11070408

**Published:** 2020-07-01

**Authors:** Yu Cao, Jie Wang, Giacinto Salvatore Germinara, Lijuan Wang, Hong Yang, Yulin Gao, Can Li

**Affiliations:** 1Guizhou Provincial Key Laboratory for Rare Animal and Economic Insect of the Mountainous Region, Department of Biology and Engineering of Environment, Guiyang University, Guiyang 550005, China; yucaosuccess@126.com (Y.C.); 17864269587@163.com (J.W.); wlj861120@163.com (L.W.); yang18798686746@163.com (H.Y.); 2Department of the Sciences of Agriculture, Food and Environment, University of Foggia, 71122 Foggia, Italy; giacinto.germinara@unifg.it; 3Key Laboratory for Biology of Plant Diseases and Insect Pests, Institute of Plant Protection, Chinese Academy of Agricultural Sciences, Beijing 100193, China

**Keywords:** olfactory responses, flower host plant, VOC, Y-tube olfactometer, six-arm olfactometer

## Abstract

*Thrips hawaiiensis* is a common thrips pest of various plant flowers with host preference. Plant volatiles provide important information for host-searching in insects. We examined the behavioral responses of *T. hawaiiensis* adults to the floral volatiles of *Gardenia jasminoides* Ellis, *Gerbera jamesonii* Bolus, *Paeonia lactiflora* Pallas, and *Rosa chinensis* Jacq. in a Y-tube olfactometer. *T. hawaiiensis* adults showed significantly different preferences to these four-flower plants, with the ranking of *G. jasminoides* > *G.*
*jamesonii* > *P. lactiflora* ≥ *R. chinensis*. Further, 29 components were identified in the volatile profiles of *G. jasminoides*, and (Z)-3-hexenyl tiglate (14.38 %), linalool (27.45 %), and (E3,E7)-4,8,12-trimethyltrideca-1,3,7,11-tetraene (24.67 %) were the most abundant. Six-arm olfactometer bioassays showed that *T. hawaiiensis* had significant positive responses to (Z)-3-hexenyl tiglate, linalool, and (E3,E7)-4,8,12-trimethyltrideca-1,3,7,11-tetraene tested at various concentrations, with the most attractive ones being 10^−3^ μL/μL, 10^−2^ μL/μL and 100 μg/μL for each compound, respectively. In pairing of these three compounds at their optimal concentrations, *T. hawaiiensis* showed the preference ranking of (Z)-3-hexenyl tiglate > linalool > (E3,E7)-4,8,12-trimethyltrideca-1,3,7,11-tetraene. Large numbers of *T. hawaiiensis* have been observed on *G. jasminoides* flowers in the field, which might be caused by the high attraction of this pest to *G. jasminoides* floral volatiles shown in the present study. Our findings shed light on the olfactory cues routing host plant searching behavior in *T. hawaiiensis*, providing important information on how *T. hawaiiensis* targets particular host plants. The high attractiveness of the main compounds (e.g., linalool, (E3,E7)-4,8,12-trimethyltrideca-1,3,7,11-tetraene, particular (Z)-3-hexenyl tiglate) identified from volatiles of *G. jasminoides* flowers may be exploited further to develop novel monitoring and control tools (e.g., lure and kill strategies) against this flower-inhabiting thrips pest.

## 1. Introduction

*Thrips hawaiiensis* Morgan (Thysanoptera: Thripidae) is a common flower-dwelling thrips pest of various horticultural plant species [1,2]. Native to the Oriental and Pacific regions, *T. hawaiiensis* is nowadays distributed in Asia, America, Africa, Australia, and Europe due to the expansion of international trade in fresh flowers, fruits, and vegetables [3,4,5,6]. In the field, *T. hawaiiensis* can attack a large number of plant species such as banana, mango, citrus, apple, tobacco, coffee, tea, horticultural plants and vegetables [6,7,8]. Therefore, it has become an important agricultural pest globally.

Chemical insecticides have always been the primary tool for *T. hawaiiensis* control, especially the high number of specific treatments applied on banana and mango crops [9,10,11]. *T. hawaiiensis* showed the potential to rapidly develop resistance to insecticides (e.g., spinetoram) under laboratory selection [9,10]. Early detection of thrips is important for growers to decide when best to apply chemical insecticides or when to alter them, which would help to limit the frequency of insecticide applications, delaying the development of insecticide-resistance. Understanding the host-location behavior in thrips would be helpful for the development of monitoring tools for their early detection.

The color, shape and volatiles associated with different plant species were considered to provide vital cues for thrips and help them to search and locate more suitable host plants [12,13,14,15,16]. Many studies have evaluated the olfactory responses of thrips, e.g., the western flower thrips *Frankliniella occidentalis* Pergande (Thysanoptera: Thripidae), indicating that both floral volatiles and non-floral odors were attractive to this pest [14,16,17,18]. Further, it was reported that rose volatile compounds allowed the design of new control strategies for western flower thrips [19]. So far little information has been mentioned on the behavioral responses of *T. hawaiiensis* to plant volatiles.

As a polyphagous flower-inhabiting thrips in China, *T. hawaiiensis* is always the dominant thrips pest on many banana and mango orchards during their flowering stages [9,10,11,20]. In addition, *T. hawaiiensis* varies in its population size on different flower host plants and vegetable crops at the flowering stage and particularly showed a preference for *Gardenia jasminoides* Ellis (Gentianales: Rubiaceae) flowers [21,22]. These results indicated that host plant flowers had a great attractiveness to *T. hawaiiensis*. To contribute to the knowledge on the role of volatile compounds in the host-searching behavior of *T. hawaiiensis*, the olfactory responses of adult insects to the odors of different flower plants including *G. jasminoides*, *Gerbera jamesonii* Bolus (Campanulales: Asteraceae), *Paeonia lactiflora* Pallas (Ranales: Ranunculaceae), and *Rosa chinensis* Jacq. (Rosales: Rosaceae) and to the main components identified from the preferred flower plant (*G. jasminoides*) were studied. These results may also help in developing new monitoring tools and other control options, which could be implemented in integrated pest management (IPM) strategies against *T. hawaiiensis*.

## 2. Materials and Methods

### 2.1. Insects and Plants

Mixed populations of *T. hawaiiensis* collected from various host plant species in the Nanming District, Guiyang area (26°34′ N, 106°42′ E) of Guizhou Province, China were used to establish a laboratory colony [21]. The independent colony was continuously reared for more than three generations on bean pods, *Phaseolus vulgaris* L. (Fabales: Leguminosae) in plastic containers [21,23]. The containers were kept in a climate-controlled room at 26 ± 1 °C, 65 ± 5% RH and a 14:10 h light:dark photoperiod.

*G. jasminoides*, *G. jamesonii*, *P. lactiflora*, and *R. chinensis* were grown in greenhouses in the nursery of Guiyang University, Guizhou Province, China. The greenhouses were maintained pest free by covering vent openings with insect-proof nettings. No pesticides were used in the whole plant growing season. Flowers at anthesis with intact petals were collected for olfactory tests and analysis of volatiles.

### 2.2. Behavioural Responses of T. hawaiiensis to Flower Volatiles

The olfactory responses of *T. hawaiiensis* were tested in a Y-tube olfactometer using the method described in Cao et al. [15,23]. Two types of two-way comparisons were made: (1) each plant flower versus clean air (CA); and (2) all possible flowers pairing. The flow rate was 300 mL/min. All bioassays were conducted between 8:00 a.m. and 6:00 p.m. in a room under 26 ± 1 °C, 65 ± 5% RH, and 1000 lux illumination conditions. For each comparison, 50 females that were 2–3 days old were tested. Thrips were starved for 4 h before the bioassay and the flower material (15.0 g) was replaced after every 10 tested individuals.

### 2.3. Collection and Analysis of Volatile Organic Compounds (VOCs)

Flower volatiles were collected and analyzed as described in Cao et al. [16]. Flower material (0.3 g) excised from a given host plant was kept in a glass bottle (200 mL) for 2 h prior to capturing the volatiles emitted using a solid-phase microextraction fiber (a ~50/30 µm DVB/CAR/PDMS StableFlex fiber). Volatiles were extracted for 40 min at 80 °C then the fiber head was quickly removed. The collected volatiles were analyzed by gas chromatography mass spectrometry (GC–MS) (HP6890/5975C, Agilent Technologies, CA, USA). The chromatographic column was a ZB-5MSI 5% phenyl-95% dimethylpolysiloxane elastic quartz capillary vessel column (30 m × 0.25 mm × 0.25 μm). The gas chromatograph was operated at an initial temperature of 40 °C for 2 min then increased at 5 °C/min to 255 °C, which was maintained for 2 min. To identify compounds, we compared the mass spectra of compounds with those in databases (Nist 2005 and Wiley 275) and their constituents were confirmed through coinjections with authentic standards.

### 2.4. Behavioural Responses of T. hawaiiensis to the Main G. jasminoides Volatile Organic Compounds (VOCs)

The VOC mixture from *G. jasminoides* was the most attractive to *T. hawaiiensis* as assessed by the Y-tube olfactometer bioassays. Since linalool, (Z)-3-hexenyl tiglate and (E3,E7)-4,8,12-trimethyltrideca-1,3,7,11-tetraene were the most abundant compounds identified in the VOC profile of *G. jasminoides*, the behavioral responses of *T. hawaiiensis* to these compounds were tested further in six-arm and Y-tube olfactometer bioassays.

### 2.5. Odour Stimuli

Mineral oil (Sigma-Aldrich, Steinheim, Germany) solutions of linalool and (Z)-3-hexenyl tiglate (Sigma-Aldrich, Germany; chemical purity 99%) (i.e., 10^−5^, 10^−4^, 10^−3^, 10^−2^ and 10^−1^ μL/μL) and (E3,E7)-4,8,12-trimethyltrideca-1,3,7,11-tetraene (Sigma-Aldrich, Germany; chemical purity 99%) (i.e., 0.1, 1, 10, 100 and 200 μg/μL) were prepared. Solutions were stored at −20 °C until the testing phase.

### 2.6. Six-arm Olfactometer Bioassays

The behavioral responses of adult *T. hawaiiensis* to different doses of linalool, (Z)-3-hexenyl tiglate, and (E3,E7)-4,8,12-trimethyltrideca-1,3,7,11-tetraene) were evaluated in a six-arm olfactometer with the method described by Liu et al. [24]. and Cao et al. [25]. Briefly, the six-arm olfactometer consisted of a central chamber with six arms, each connected to a glass tube that projected outwards at an equidistance, with equal angles (60°) between pairs of tubes. Each arm was connected through Teflon tubing to a glass vessel containing a test or control stimulus. For each experiment, equal volumes (25 µL) of each of the five solutions of one compound and mineral oil (used as the control), absorbed onto a filter paper disk (1.0-cm diameter), were used as test and control stimuli, respectively. The airflow was set at 200 mL/min to drive the odor source to thrips. *Thrips hawaiiensis* (2–3 days old females) were starved for 4 h and introduced in groups (200 individuals per group) into the central chamber with a fine camel hair brush. Within 20 min, insects that entered one arm of the olfactometer were counted as having made a choice for a particular odor, while thrips that did not enter any arm were considered non-responders. After each test, the olfactometer was cleaned, dried and the arms were rotated (60°). Each bioassay was replicated six times between 9:00 am and 6:00 pm. In order to eliminate any light bias, a 25-W light was placed in the center 60 cm above the chamber.

### 2.7. Y-Tube Bioassays

In the six-arm olfactometer bioassays, linalool, (Z)-3-hexenyl tiglate, and (E3,E7)-4,8,12-trimethyltrideca-1,3,7,11-tetraene showed the highest attractiveness to *T. hawaiiensis* at the concentration of 10^−2^ μL/μL, 10^−3^ μL/μL and 100 μg/μL, respectively. Therefore, the attractant power of the three compounds at their optimal concentrations were compared in further Y-tube bioassays as described above.

### 2.8. Statistical Analyses

All statistical analyses were performed using SPSS 18.0 for Windows (SPSS Inc., Chicago, IL, USA) [26]. The null hypothesis that *T. hawaiiensis* adults showed no preference for either Y-tube arm (a response equal to 50:50) was analyzed using a chi-square goodness-of-fit test [27]. The number of thrips found in the different arms of the six-arm olfactometer were subjected to Friedman two-way ANOVA by ranks and in the case of significance (*p* < 0.05) the Wilcoxon signed ranks test was used for separation of means [28].

## 3. Results

### 3.1. Behavioural Responses of T. hawaiiensis to Flower Volatiles

When *T. hawaiiensis* adults were presented with different flower volatiles *versus* clean air (CA), they showed significant preferences for *G. jasminoides* (*χ*^2^ = 25.13, *df* = 1, *p <* 0.01), *G. jamesonii* (*χ*^2^ = 16.20, *df* = 1, *p <* 0.01), *P. lactiflora* (*χ*^2^ = 5.82, *df* = 1, *p =* 0.016) and *R. chinensis* (*χ*^2^ = 5.00, *df* = 1, *p* = 0.025) (Figure 1).

*Thrips hawaiiensis* also showed significant preferences among the flowers when presented with choices. *Gardenia jasminoides* was more attractive to *T. hawaiiensis* than *G. jamesonii* (*χ*^2^ = 4.79, *df* = 1, *p* = 0.029), *P. lactiflora* (*χ*^2^ = 6.15, *df* = 1, *p* = 0.013) or *R. chinensis* (*χ*^2^ = 8.02, *df* = 1, *p =* 0.005). *Gardenia jasminoides* was more attractive than *P. lactiflora* (*χ*^2^ = 4.26, *df* = 1, *p* = 0.039) or *R. chinensis* (*χ*^2^ = 5.23, *df* = 1, *p =* 0.022). However, *T. hawaiiensis* showed no significant preference between *P. lactiflora* and *R. chinensis* (*χ*^2^ = 2.69, *df* = 1, *p* = 0.10) (Figure 1).

### 3.2. Analysis of G. jasminoides Volatiles

Twenty-nine components were identified in the volatiles from *G. jasminoides* (Table 1). The component with the highest relative content was linalool (27.45%), followed by (E3,E7)-4,8,12-trimethyltrideca-1,3,7,11-tetraene (24.67%), (Z)-3-hexenyl tiglate (14.38%), and then jasmine lactone (6.93%). There was no other component greater than 5% identified from the flower volatiles of *G. jasminoides*.

### 3.3. Behavioural Responses of T. hawaiiensis to the Main Components of G. jasminoides VOCs

#### 3.3.1. Six-Arm Olfactometer Bioassays

In these experiments, the number of insects that entered the control arm connected to the vessel with mineral oil were significantly lower than those of insects found in the arms with the different doses of linalool (Friedman test: *χ*^2^ = 30.00, *df* = 5, *p* < 0.001, Wilcoxon tests: *p* = 0.026–0.028), (Z)-3-hexenyl tiglate (Friedman test: *χ*^2^ = 29.88, *df* = 5, *p* < 0.001, Wilcoxon tests: *p* = 0.026–0.027), or (E3,E7)-4,8,12-trimethyltrideca-1,3,7,11-tetraene (Friedman test: *χ*^2^ = 29.88, *df* = 5, *p* < 0.001, Wilcoxon tests: *p* = 0.026–0.028) (Figure 2).

There were relatively significant differences in attraction among all doses of linalool (Wilcoxon tests: *p* = 0.026–0.028), (Z)-3-hexenyl tiglate (Wilcoxon tests: *p* = 0.026–0.038), and (E3,E7)-4,8,12-trimethyltrideca-1,3,7,11-tetraene (Friedman test: *χ*^2^ = 29.88, *df* = 5, *p* < 0.001, Wilcoxon tests: *p* = 0.026–0.043) (Figure 3). Significantly more insects entered the arm connected to the vessel that contained 0.25 μL of linalool or 0.025 μL of (Z)-3-hexenyl tiglate or 0.25 mg of (E3,E7)-4,8,12-trimethyltrideca-1,3,7,11-tetraene over the arm with the other doses.

#### 3.3.2. Y-Tube Olfactometer Bioassays

In the six-arm olfactometer bioassays, 10^-2^ μL/μL for linalool, 10^-3^ μL/μL for (Z)-3-hexenyl tiglate and 100 μg/μL for (E3,E7)-4,8,12-trimethyltrideca-1,3,7,11-tetraene, respectively, were the most attractive concentrations to *T. hawaiiensis*. Thus, the behavioral responses of *T. hawaiiensis* to these compounds at their optimal doses were further compared in a Y-tube olfactometer (Figure 3). Significantly higher numbers of *T. hawaiiensis* were found to prefer (Z)-3-hexenyl tiglate (*χ*^2^ = 27.00, *df* = 1, *p* < 0.01), linalool (*χ*^2^ = 25.13, *df* = 1, *p* < 0.01), and (E3,E7)-4,8,12-trimethyltrideca-1,3,7,11-tetraene (*χ*^2^ = 19.57, *df* = 1, *p* < 0.01) as compared to clean air (Figure 3).

When these three compounds were compared with each other, *T. hawaiiensis* significantly preferred (Z)-3-hexenyl tiglate to linalool (*χ*^2^ = 5.33, *df* = 1, *p* = 0.021), (Z)-3-hexenyl tiglate to (E3,E7)-4,8,12-trimethyltrideca-1,3,7,11-tetraene (*χ*^2^ = 5.57, *df* = 1, *p* = 0.018), and linalool to (E3,E7)-4,8,12-trimethyltrideca-1,3,7,11-tetraene (*χ*^2^ = 4.46, *df* = 1, *p* = 0.035) (Figure 3).

## 4. Discussion

Volatile compounds emitted from plants are important cues in the host selection process of phytophagous insects. In a natural plant community, different plant species emit different qualitative and quantitative blends of VOCs, which guide insects to discriminate and locate their host plants [29,30]. For flower thrips pests, a great number of studies have documented that *F. occidentalis* exhibited significantly positive responses to the volatiles of host plants [13,14,15,17,18], and displayed a significant preference for specific host olfactory cues [16,23]. Therefore, based on the bioactive compounds identified from the host plant volatiles, lures were developed and applied for the monitoring and control of *F. occidentalis* [19,31,32,33,34]. In the present study, *T. hawaiiensis* were significantly attracted to the volatiles from *G. jasminoides*, *G. jamesonii*, *P. lactiflora*, and *R. chinensis*, and showed olfactory preferences with *G. jasminoides* > *G. jamesonii* > *P. lactiflora* ≥ *R. chinensis*. This is consistent with our previous field observations during which large numbers of *T. hawaiiensis* were found in *G. jasminoides* flowers [21]. Furthermore, the nutritional composition of *G. jasminoides* flowers could adequately satisfy the nutritional needs of *T. hawaiiensis* which could have a faster population development when fed on these flowers [35].

GC-MS analysis highlighted the presence of 29 compounds in the VOC profile of *G. jasminoides* flowers, among which the most abundant compounds, linalool (27.45 %), (E3,E7)-4,8,12-trimethyltrideca-1,3,7,11-tetraene (24.67 %), and (Z)-3-hexenyl tiglate (14.38 %), were considered to be related to the typical floral odor of this plant species [36]. Different concentrations of linalool (10^-5^ ~ 10^-1^ μL/μL), (Z)-3-hexenyl tiglate (10^-5^ ~ 10^-1^ μL/μL), and (E3,E7)-4,8,12-trimethyltrideca-1,3,7,11-tetraene (0.1 ~ 200 μg/μL) were all attractive to *T. hawaiiensis*, but the 10^-2^ μL/μL, 10^-3^ μL/μL, and 100 μg/μL concentrations, respectively, were the most attractive ones. Further, regarding these three compounds at their optimal attractive concentration, *T. hawaiiensis* had different olfactory preferences with (Z)-3-hexenyl tiglate > linalool > (E3,E7)-4,8,12-trimethyltrideca-1,3,7,11-tetraene. Therefore, (Z)-3-hexenyl tiglate seems to have greater potential for the development of a new attractant lure for *T. hawaiiensis* monitoring and control. The behavioral responses of insects are influenced not only by individual compounds and concentrations but also by their ratios in mixtures [37,38,39]. So, the possible additive or synergistic effects among (Z)-3-hexenyl tiglate and the other attractive compounds identified in this study may require to be further evaluated through massive behavioral bioassays testing different doses and combinations of these VOCs. Moreover, field-trapping tests should be conducted to confirm their practically attractive effects [18,19,30,37].

Pollens were considered to be an important factor involved in the preference of thrips for host plant flowers [40,41,42], as *F. occidentalis* was attracted to the compound of (*S*)-verbenone identified from the volatiles of pine pollen [18,33]. However, pollens of *G. jasminoides* flowers were not involved in *T. hawaiiensis*’ olfactory responses in this study. Besides, the host preference of different thrips species were related to the color, shape, nutritional conditions or other physicochemical characteristics of host plants [13,15,17,43,44,45]. Thus, more related physicochemical characteristics that may influence the behavioral responses of *T. hawaiiensis* should be studied to comprehensively understand the mechanism of host selection among different flower plants.

Host plant volatiles not only attract phytophagous pests but also their natural enemies [46,47,48] Although predators seem to be more attracted by herbivore-induced volatiles, as reported on the significant preferences of the predator *Neoseiulus cucumeris* Oudemans (Acari: Phytoseiidae) to the volatiles from vegetable hosts infested by *F. occidentalis* or *Thrips tabaci* Lindeman (Lindeman) (Thysanoptera: Thripidae) [47,49], data that do not support such a specificity have been reported [50,51,52,53]. In this framework, it is imperative to study the attractiveness of VOCs from healthy and infested flowers to natural enemies of *T. hawaiiensis*, e.g., *Orius sauteri* Poppius (Heteroptera: Anthocoridae) [54].

## 5. Conclusions

Females of *T. hawaiiensis* exhibited a higher olfactory preference for the VOCs of *G. jasminoides* flowers over a range of four flower plants. Moreover, insects were significantly attracted by different concentrations of the three main components of *G. jasminoides* flower VOCs with (Z)-3-hexenyl tiglate being the most attractive. Overall, this study clearly demonstrated that plant volatiles are involved in host-plant selection by *T. hawaiiensis* even if, to comprehensively understand this mechanism, possible interactions among chemical cues and other physicochemical characteristics remain to be investigated. The kairomonal activity of (Z)-3-hexenyl tiglate, linalool, and (E3,E7)-4,8,12-trimethyltrideca-1,3,7,11-tetraene to *T. hawaiiensis* females found in this study provides a basis for further electrophysiological, behavioral, and field-trapping experiments to develop semiochemically-based monitoring tools and direct control options for this pest.

## Figures and Tables

**Figure 1 insects-11-00408-f001:**
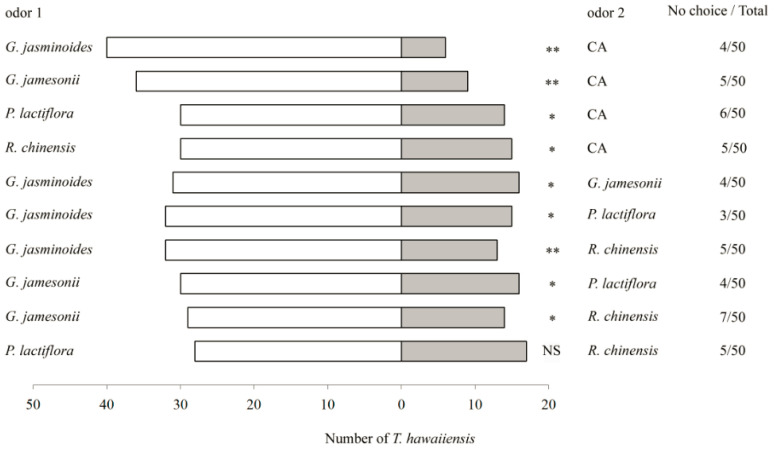
Behavioral responses of *T. hawaiiensis* to the volatiles from different flowers. Asterisks indicate highly significant (** *p* < 0.01) and significant (* *p* < 0.05) differences in the selectivity of *T. hawaiiensis* between two odors by *χ*^2^ test. NS indicates no significant differences (*p* > 0.05) in the selectivity of *T. hawaiiensis* between two odors.

**Figure 2 insects-11-00408-f002:**
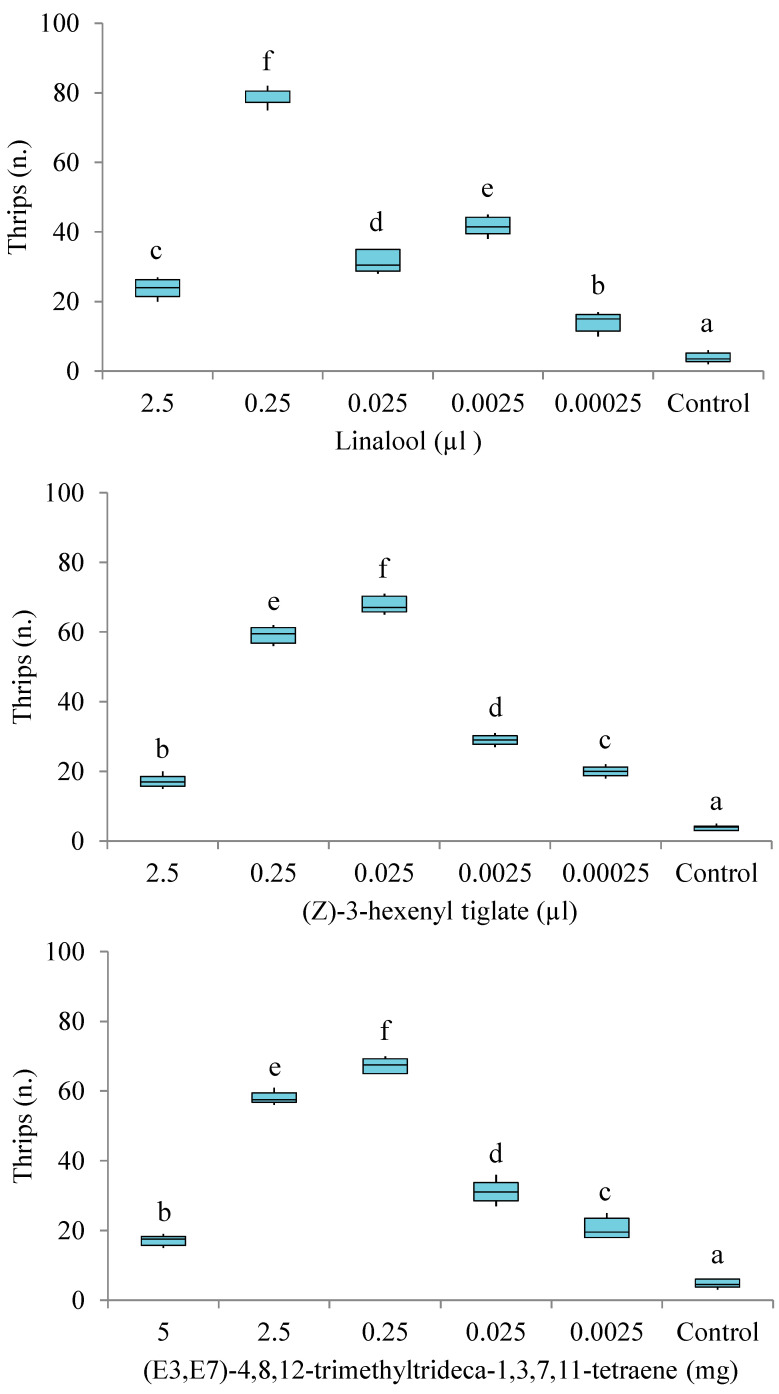
Olfactory responses of *T. hawaiiensis* to different doses of linalool, (Z)-3-hexenyl tiglate and (E3,E7)-4,8,12-trimethyltrideca-1,3,7,11-tetraene in a six-arm olfactometer. Control was mineral oil. Each box plot represents the median and its range of dispersion (lower and upper quartiles and outliers). Above each box plot, different letters indicate significant differences (Wilcoxon test, *p* < 0.05).

**Figure 3 insects-11-00408-f003:**
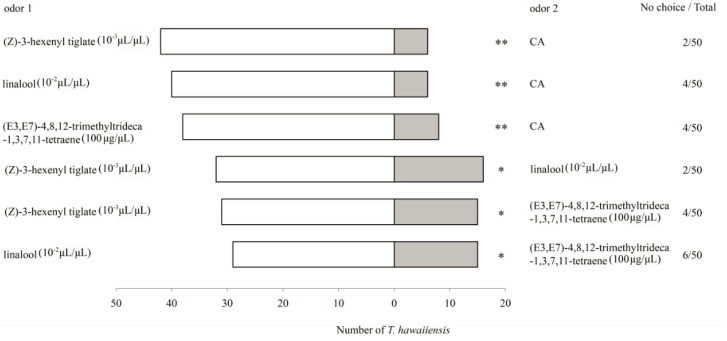
Behavioral responses of *T. hawaiiensis* to the main *Ga. jasminoides* volatiles. Asterisks indicate highly significant (** *p* < 0.01) and significant (* *p* < 0.05) differences in the selectivity of *T. hawaiiensis* between two odors by *χ*^2^ test.

**Table 1 insects-11-00408-t001:** Volatile components of *G. jasminoides* flower.

Number	Compounds	Molecular Formula	Molecular Weight	Content (%)
1	Methyl tiglate	C_6_H_10_O_2_	114	0.39
2	(Z)-3-Hexen-1-ol	C_6_H_12_O	100	3.59
3	β-Myrcene	C_10_H_16_	136	0.15
4	(Z)-3-Hexenyl acetate	C_8_H_14_O_2_	142	0.18
5	(E)-Ocimene	C_10_H_16_	136	1.38
6	γ-Caprolactone	C_6_H_10_O_2_	114	0.10
7	trans-Linalool oxide	C_10_H_18_O_2_	170	0.32
8	Methyl benzoate	C_8_H_8_O_2_	136	1.64
9	Linalool	C_10_H_18_O	154	27.45
10	(Z)-3-hexenyl iso-butyrate	C_10_H_18_O_2_	134	0.22
11	Benzyl acetate	C_9_H_10_O_2_	150	0.25
12	(Z)-3-hexenyl butanoate	C_10_H_18_O_2_	170	0.38
13	Methyl salicylate	C_8_H_8_O_3_	152	0.20
14	(Z)-3-hexenyl 2-methylbutanoate	C_11_H_20_O_2_	184	1.28
15	Hexyl 2-Methylbutyrate	C_11_H_22_O_2_	186	0.61
16	Benzyl propionate	C_10_H_12_O_2_	164	0.12
17	(Z)-3-hexenyl tiglate	C_11_H_18_O_2_	182	14.38
18	Hexyl tiglate	C_11_H_20_O_2_	184	4.01
19	(Z)-3-hexenyl hexanoate	C_12_H_22_O_2_	198	0.11
20	Isoamyl benzoate	C_12_H_16_O_2_	192	0.28
21	γ-Decanolactone	C_10_H_18_O_2_	170	0.18
22	Jasmine lactone	C_10_H_16_O_2_	168	6.93
23	Benzyl tiglate	C_12_H_14_O_2_	190	0.23
24	α-Farnesene	C_15_H_24_	204	3.58
25	Octyl (E)-2-methylbut-2-enoate	C_13_H_24_O_2_	212	0.58
26	E-Nerolidol	C_15_H_26_O	222	0.27
27	(Z)-3-Hexenyl phenylacetate	C_14_H_18_O_2_	218	0.41
28	(E3,E7)-4,8,12-trimethyltrideca-1,3,7,11-tetraene	C_16_H_26_	218	24.67
29	Geranyl tiglate	C_15_H_24_O_2_	236	0.41

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
