# Peer review of "Behavioral Responses of Thrips hawaiiensis (Thysanoptera: Thripidae) to Volatile Compounds Identified from Gardenia jasminoides Ellis (Gentianales: Rubiaceae)"

_insects, 2020, doi:10.3390/insects11070408_

Round 1

Reviewer 1 Report

Comments from Reviewer:

The authors focusses on the behavioural response of T. hawaiiensis to the host plant G. jasminoides and identified the attractive volatile compounds. Some of the compounds were individually tested of its activity with Y-tube olfactometer and those were active compared with clean air. One important thing the authors did not perform but just discussed briefly is the effect of mixing several identified compounds. Also, the bioassay with Y-tube is just a preference test, they cannot conclude that these identified chemicals are active enough to explain the behaviour in the field. They should test it with a wind tunnel and conclude its attractiveness.  In Conclusion, authors discussed the volatiles for thrips (kairomone) and also referred to the HIPVs for predators (synomone). This discussion is very confusing, since authors mentioned that the goal of this project is to develop novel monitoring and control tool against this pest. Attracting natural enemies is totally different phenomenon and should not be mixed together with kairomone for pest insects. Usually HIPVs are released from plants that was induced by herbivore feeding, and are specific group of chemicals. Those chemicals are different from the attractive volatiles for thrips.

Minor points:

Authors mistook the chemicals’ names so often. They mixed together with the use of (Z) and cis, (E) and trans. Also, they mixed upper case and lower case in terms of chemical name, e.g., Hexenyl and hexenyl. There is no uniformity in the text and also Table 1.

L.158: ‘(E)-Hex-3-enyl (E)-2-methylbut-2-enoate’ is the formal name of (Z)-3-hexenyl tiglate? ‘(Z)-Hex-3-enyl (Z)-2-methylbut-2-enoate’ may be a right name? You should refer to that these two compounds are the same compound somewhere. If you use these two names without explanation, readers will be confused.

L.371: ‘Satoshi, T.; Takeshi, S.’ should be ‘Tatemoto, S.; Shimoda, T.’ (family and first names are opposite).

I can recommend to authors that they should check all the chemical names and resubmit to the journal.

Author Response

The authors focusses on the behavioural response of T. hawaiiensis to the host plant G. jasminoides and identified the attractive volatile compounds. Some of the compounds were individually tested of its activity with Y-tube olfactometer and those were active compared with clean air. One important thing the authors did not perform but just discussed briefly is the effect of mixing several identified compounds. Also, the bioassay with Y-tube is just a preference test, they cannot conclude that these identified chemicals are active enough to explain the behaviour in the field. They should test it with a wind tunnel and conclude its attractiveness. In Conclusion, authors discussed the volatiles for thrips (kairomone) and also referred to the HIPVs for predators (synomone). This discussion is very confusing, since authors mentioned that the goal of this project is to develop novel monitoring and control tool against this pest. Attracting natural enemies is totally different phenomenon and should not be mixed together with kairomone for pest insects. Usually HIPVs are released from plants that was induced by herbivore feeding, and are specific group of chemicals. Those chemicals are different from the attractive volatiles for thrips.

Authors’ response: Many thanks to the referee for this comments. Accordingly, we have revised the discussion and conclusion sections but taking into account that carnivores are not always attracted by HPVs. There are many scientific evidences that also compounds constitutively emitted by planta that can attract predators or parasitoids (see references number 50-53).

As also reported in the reference n. 46 (Dicke, 1999 M. Are herbivore-induced plant volatiles reliable indicators of herbivore identity to foraging carnivorous arthropods? Entomol Exp Appl. 1999, 92, 131–142) “Studies that support a specificity of herbivore-induced plant volatiles (e.g., Sabelis & van de Baan,1983; Takabayashi et al., 1995; Du et al., 1996; Powellet al., 1998; de Moraes et al., 1998) as well as data that do not support such a specificity (e.g., Turlings et al., 1993b, 1998; Mattiacci & Dicke, 1995; Röse et al., 1998) have been presented for different plant-herbivore-carnivore systems”.

In addition, we have improved the discussion and conclusion. Lines 226-229, 252-272.

Minor points:

Authors mistook the chemicals’ names so often. They mixed together with the use of (Z) and cis, (E) and trans. Also, they mixed upper case and lower case in terms of chemical name, e.g., Hexenyl and hexenyl. There is no uniformity in the text and also Table 1.

Authors’ response: We have checked all the chemicals’ names and unified their names in the text.

L.158: ‘(E)-Hex-3-enyl (E)-2-methylbut-2-enoate’ is the formal name of (Z)-3-hexenyl tiglate? ‘(Z)-Hex-3-enyl (Z)-2-methylbut-2-enoate’ may be a right name? You should refer to that these two compounds are the same compound somewhere. If you use these two names without explanation, readers will be confused.

Authors’ response: Sorry for our carelessness, (Z)-Hex-3-enyl (Z)-2-methylbut-2-enoate is the right name for (Z)-3-hexenyl tiglate,and the  accurate name for (Z)-3-hexenyl tiglate should be cis-3-Hexenyl tiglate, which was only used in the revised manuscript maintain the uniformity.

L.371: ‘Satoshi, T.; Takeshi, S.’ should be ‘Tatemoto, S.; Shimoda, T.’ (family and first names are opposite).

Authors’ response: Corrected. Line 403.

I can recommend to authors that they should check all the chemical names and resubmit to the journal.

Authors’ response: Thank you for your suggestion, we have done it.

Many thanks,

     Yu Cao, Yulin Gao, Can Li, et al.

Reviewer 2 Report

The manuscript “Behavioral responses of Thrips hawaiiensis (Thysanoptera: Thripidae) to volatile compounds identified from Gardenia jasminoides (Gentianales: Rubiaceae)” evaluated the attractiveness of the olfactory cues from various flowers, as well as the attractiveness of the most abundant VOCs from the preferred flower source toward female specimens of the thrips pest T. hawaiiensis. The manuscript is well conceived and the experimental set up is generally well designed. However, some more info are needed to clarify the methods and the discussions need to be improved and expanded (see specific points below). Lastly, the language is not always correct, either grammatically as well as for the incorrect use of the terminology. Thus, I suggest revising the whole text to improve it (in the list below I provided some examples).  

SPECIFIC POINTS:

LN 17 “with different host preference.” Unclear.

LN 30 “dominant numbers of T. hawaiiensis”. Rephrase.

LN 31 “which was reasonably related” Who/what was related?

LN 62 “a dominant thrips pest”. Or this “THE dominant” pest species or it “a KEY” species.

LN 70 “most preferred” Incorrect. Just “preferred”.

LN 87-92 Please add air flow information.

LN 88 “each flower”. Grammatically incorrect. How many flowers per type were used in the olfactometer? One?

LN 91 “T. hawaiiensis females–2-3-day-old”. T. hawaiiensis females 2-3-day-old

LN 94-8 I understand that the information about the methods are described in Cao et al, but I believe that some info are needed by the readers also in the present manuscript. i.e. “solid-phase microextraction fiber”. Which fiber? How long was the exposition time of the fiber (the whole 2 hours or there were some incubation)? GC-MS analyses: please add at least the column and the program used.

LN 204 “and performed significantly different host preferences via olfactory cues”. Bad English. I suggest “and displayed significant preference for specific host olfactory cues”.

LN 208 “different preference”. A preference cannot be different. The attractiveness can.

LN 209 “This may partially explain why T. hawaiiensis showed a strong preference to G. jasminoides flowers in the field [21, 35].” I would expand this part of the discussion.

LN 211 “VOCs profile”. Change in “VOC profile”.

LN 217 attractant is not an adjective, is a noun. The adjective is “attractive”.

LN 229 “However, pollens of G. jasminoides flowers were not involved in T. hawaiiensis’ olfactory responses in this study.” I cannot understand how the authors excluded it. Were the tested flowers deprived of pollen? Please explain.

LN 230 “Besides, it was believed that the host preference of different thrips species were related to the color, shape, nutritional conditions or other physicochemical characteristics of host plants [13, 15, 17, 43-45].” Why it WAS believed? It is not true now?

LN 242 T. tabaci. First mention. Please use extended name and add authorship and classification.

LN 244 Orius sauteri. First mention. Please add authorship and classification.

Author Response

The manuscript “Behavioral responses of Thrips hawaiiensis (Thysanoptera: Thripidae) to volatile compounds identified from Gardenia jasminoides (Gentianales: Rubiaceae)” evaluated the attractiveness of the olfactory cues from various flowers, as well as the attractiveness of the most abundant VOCs from the preferred flower source toward female specimens of the thrips pest T. hawaiiensis. The manuscript is well conceived and the experimental set up is generally well designed. However, some more info are needed to clarify the methods and the discussions need to be improved and expanded (see specific points below). Lastly, the language is not always correct, either grammatically as well as for the incorrect use of the terminology. Thus, I suggest revising the whole text to improve it (in the list below I provided some examples). 

Authors’ response: Thank you for the opportunity to revise our manuscript again. We appreciate the suggestions made by the reviewer to improve our paper. Detailed responses to the comments and suggestions are provided below:

SPECIFIC POINTS:

LN 17 “with different host preference.” Unclear.

Authors’ response: We have improved the writting. Line 17.

LN 30 “dominant numbers of T. hawaiiensis”. Rephrase.

Authors’ response: Rephrased. Line 30.

LN 31 “which was reasonably related” Who/what was related?

Authors’ response: We have improved the writting. Line 31.

LN 62 “a dominant thrips pest”. Or this “THE dominant” pest species or it “a KEY” species.

Authors’ response: Corrected. Line 65.

LN 70 “most preferred” Incorrect. Just “preferred”.

Authors’ response: Corrected. Line 73.

LN 87-92 Please add air flow information.

Authors’ response: Added. Line 92.

LN 88 “each flower”. Grammatically incorrect. How many flowers per type were used in the olfactometer? One?

Authors’ response: Sorry for the ambiguous expression, we have improved the writting. “each flower” means “each type flower from each tested host plant”. Line 91.

LN 91 “T. hawaiiensis females–2-3-day-old”. T. hawaiiensis females 2-3-day-old

Authors’ response: Corrected. Line 94.

LN 94-8 I understand that the information about the methods are described in Cao et al, but I believe that some info are needed by the readers also in the present manuscript. i.e. “solid-phase microextraction fiber”. Which fiber? How long was the exposition time of the fiber (the whole 2 hours or there were some incubation)? GC-MS analyses: please add at least the column and the program used.

Authors’ response: Thank you for your suggestion, we have added the necessary information. Lines 100-102, 103-107.

LN 204 “and performed significantly different host preferences via olfactory cues”. Bad English. I suggest “and displayed significant preference for specific host olfactory cues”.

Authors’ response: Thank you for your suggestion, we have corrected it. Lines 220-221.

LN 208 “different preference”. A preference cannot be different. The attractiveness can.

Authors’ response: We have improved the writing. Line 225.

LN 209 “This may partially explain why T. hawaiiensis showed a strong preference to G. jasminoides flowers in the field [21, 35].” I would expand this part of the discussion.

Authors’ response: Thank you for your suggestion, we have improved the discussion. Lines 226-229.

LN 211 “VOCs profile”. Change in “VOC profile”.

Authors’ response: Done. Line 230.

LN 217 attractant is not an adjective, is a noun. The adjective is “attractive”.

Authors’ response: Corrected. Line 236.

LN 229 “However, pollens of G. jasminoides flowers were not involved in T. hawaiiensis’ olfactory responses in this study.” I can not understand how the authors excluded it. Were the tested flowers deprived of pollen? Please explain.

Authors’ response: Yes, just flower petals, which were deprived of pollen, were used in our tests. Therefore, pollens of G. jasminoides flowers were not involved in T. hawaiiensis’ olfactory responses.

LN 230 “Besides, it was believed that the host preference of different thrips species were related to the color, shape, nutritional conditions or other physicochemical characteristics of host plants [13, 15, 17, 43-45].” Why it WAS believed? It is not true now?

Authors’ response: We have deleted it. Line 250.

LN 242 T. tabaci. First mention. Please use extended name and add authorship and classification.

Authors’ response: Corrected. Lines 258.

LN 244 Orius sauteri. First mention. Please add authorship and classification.

Authors’ response: Corrected. Line 261.

Many thanks,

     Yu Cao, Yulin Gao, Can Li, et al.

Reviewer 3 Report

The authors demonstrated host-flower preference of a thrips pest Thrips hawaiiensis using flower derived volatiles and identified attractive chemical compound from volatiles of Gardenia jasminoides. The experimental design is simple, the results support the conclusions and the manuscript is well written.  I have only minor comments.

L48: Morgan > (Morgan)

(The author name should be in brackets if the species is transferred to a genus other than the one in which it was originally described.)

L43: horticultural > Please delete an underline

L47: [6, 7-8] > [6-8]

L91: Please explain why the authors used only females

L205: bioactive > Please delete an underline

L371: Satoshi, T.; Takeshi, S. > Takemoto, S.; Shimoda, T.

("Satoshi" and "Shimoda" are their first name)

Author Response

The authors demonstrated host-flower preference of a thrips pest Thrips hawaiiensis using flower derived volatiles and identified attractive chemical compound from volatiles of Gardenia jasminoides. The experimental design is simple, the results support the conclusions and the manuscript is well written. I have only minor comments.

Authors’ response: Thanks for the opportunity to revise our manuscript. We appreciate the positive evaluation on our work.

L48: Morgan > (Morgan) (The author name should be in brackets if the species is transferred to a genus other than the one in which it was originally described.)

Authors’ response: Corrected. Line 43.

L43: horticultural > Please delete an underline

Authors’ response: Deleted. Line 44.

L47: [6, 7-8] > [6-8]

Authors’ response: Corrected. Line 48.

L91: Please explain why the authors used only females

Authors’ response: In our previous study, we found that female T. hawaiiensis were better able to recognize host plants by olfactory means compared with male thrips [1]. Therefore we used only females in this study.

[1] Cao Y, Meng YL, Yang H, Li J, Zhang GZ, Wang YW, Li C. Preliminary study on the behavioral responses of Thrips hawaiiensis to volatiles of different flowers. Journal of Henan Agricultural Science, 2020, accepted. (in Chinese with English abstract)

L205: bioactive > Please delete an underline

Authors’ response: Deleted. Line 222.

L371: Satoshi, T.; Takeshi, S. > Takemoto, S.; Shimoda, T. ("Satoshi" and "Shimoda" are their first name)

Authors’ response: Corrected. Line 403.

Many thanks,

     Yu Cao, Yulin Gao, Can Li, et al.

Round 2

Reviewer 1 Report

I could still find the lack of uniformity.  Please look at the following line. If you use 'Hexenyl', you should use 'Linalool'. You can use (Z)- instead of cis-, since you already use '(3E,7E)' in all the text.

 L.238: cis-3-Hexenyl tiglate > linalool >

If I were you, I will write as follows:

 L.238: (Z)-3-hexenyl tiglate > linalool >

This way might be commonly used in other journals like J. Chem. Ecol. Please look at all the text and unify the usage of chemical names.

Author Response

I could still find the lack of uniformity.  Please look at the following line. If you use 'Hexenyl', you should use 'Linalool'. You can use (Z)- instead of cis-, since you already use '(3E,7E)' in all the text.

 L.238: cis-3-Hexenyl tiglate > linalool >

If I were you, I will write as follows:

 L.238: (Z)-3-hexenyl tiglate > linalool >

This way might be commonly used in other journals like J. Chem. Ecol. Please look at all the text and unify the usage of chemical names.

Authors’ response: We have checked all the chemicals’ names and unified their names in the text. Please see the attachment.

Many thanks,

     Yu Cao, Yulin Gao, Can Li, et al.
